# Comparative Whole-Genome Sequence Analyses of Fusarium Wilt Pathogen (*Foc* R1, STR4 and TR4) Infecting Cavendish (AAA) Bananas in India, with a Special Emphasis on Pathogenicity Mechanisms

**DOI:** 10.3390/jof7090717

**Published:** 2021-08-31

**Authors:** Thangavelu Raman, Esack Edwin Raj, Gopi Muthukathan, Murugan Loganathan, Pushpakanth Periyasamy, Marimuthu Natesh, Prabaharan Manivasakan, Sharmila Kotteeswaran, Sasikala Rajendran, Uma Subbaraya

**Affiliations:** 1Plant Pathology Division, ICAR-National Research Centre for Banana, Tiruchirappalli, Tamil Nadu 620102, India; edwinrht@gmail.com (E.E.R.); gopimusa@gmail.com (G.M.); logumuruga@gmail.com (M.L.); Lalitha_pushpakanth@yahoo.com (P.P.); marikowsh@gmail.com (M.N.); praba.nrcb@gmail.com (P.M.); sharmilakotteeswaran@gmail.com (S.K.); sasmbi06@gmail.com (S.R.); umabinit@yahoo.co.in (U.S.); 2Research and Development Division, MIRO Forestry SL Ltd., Mile 91, Tonkolili District, Northern Provenance P.O. Box GP20200, Sierra Leone

**Keywords:** banana, fusarium wilt, pathogenicity genes, SIX genes, whole-genome sequencing

## Abstract

Fusarium wilt is caused by the fungus *Fusarium oxysporum* f. sp. *cubense* (*Foc*) and is the most serious disease affecting bananas (*Musa* spp.). The fungus is classified into *Foc* race 1 (R1), *Foc* race 2, and *Foc* race 4 based on host specificity. As the rate of spread and the ranges of the devastation of the *Foc* races exceed the centre of the banana’s origin, even in non-targeted cultivars, there is a possibility of variation in virulence-associated genes. Therefore, the present study investigates the genome assembly of *Foc* races that infect the Cavendish (AAA) banana group in India, specifically those of the vegetative compatibility group (VCG) 0124 (race 1), 0120 (subtropical race 4), and 01213/16 (tropical race 4). While comparing the general features of the genome sequences (e.g., RNAs, GO, SNPs, and InDels), the study also looked at transposable elements, phylogenetic relationships, and virulence-associated effector genes, and sought insights into race-specific molecular mechanisms of infection based on the presence of unique genes. The results of the analyses revealed variations in the organisation of genome assembly and virulence-associated genes, specifically secreted in xylem (SIX) genes, when compared to their respective reference genomes. The findings contributed to a better understanding of Indian *Foc* genomes, which will aid in the development of effective Fusarium wilt management techniques for various Foc VCGs in India and beyond.

## 1. Introduction

India is the world’s largest producer of the fifth most vital horticultural fruit crop, the banana (*Musa* spp.) [1]. Although several diseases wreak havoc on banana production around the world, Fusarium wilt, caused by a soil-borne fungus *Fusarium oxysporum* f. sp. *cubense* (*Foc*), has received more attention due to its scientific and economic significance [2]. In India, production losses caused by the devastating wilt disease severely harm the national economy as well as the livelihoods of banana farmers [3]. To overcome economic hardship, rapid, accurate, and early diagnostic procedures, as well as the effective implementation of quarantine/control measures, are critical for the effective containment of the disease [4].

The first record of Fusarium wilt disease was reported in Java in the 18th century, in the most widely cultivated banana strain at the time, Gros Michel [5]. At present, the disease has spread to many banana-growing countries, including a recent outbreak of the disease in the Cavendish banana in India involving *Foc* Race 1 (VCG0124), *Foc* subtropical race 4 (STR4, VCG01220) [6] and *Foc* tropical race 4 (TR4, VCG01213/16) [7,8]. The dangers of global banana monoculture were first highlighted by the spread of the harmful strain *Foc* R1 to the majority of the world’s banana producing countries, destroying the Gros Michel banana [9]. The recent outbreak of Fusarium wilt in the Cavendish banana is caused by a unique vegetative compatibility group, VCG01213/16, known as *Foc* TR4. This is the most destructive race because of its much broader host range (Cavendish banana and all cultivars sensitive to the *Foc* R1 and *Foc* R2) and widespread distribution far from its origin [7,10]. These *Foc* races have resulted in significant economic losses to commercial and subsistence banana producing areas of Australia, Cambodia, China, Colombia, India, Indonesia, Israel, Jordan, Laos, Lebanon, Malaysia, Mozambique, Myanmar, Oman, Pakistan, Taiwan, Thailand, the Philippines, Turkey, and Vietnam [3].

*Foc* produces dormant resting spores known as chlamydospores, which allow the pathogen to survive without a host for at least a few decades, preventing any susceptible banana cultivars from being grown in disease-infected fields in the future [5]. The disease cycle begins with an infection of the root system and colonises the vascular tissue, induces a reddish-brown discolouration in the rhizome and pseudostem, disrupts water and nutrient transportation, resulting in chlorosis, pseudostem splitting, and wilting [2,5], and eventually, results in plant death before fruit bunches are produced, resulting in a significant reduction in crop yield [11]. The disease incidence varies depending on banana cultivars/genotypes, environment, and *Foc* inoculum levels, but it can result in total crop loss in heavily affected fields [12]. In recent years, *Foc* TR4 and R1 have spread rapidly into Cavendish bananas in India and other regions, posing a threat to the global banana industry and drawing international attention to the future food security of banana-producing countries. Although the incidence of Fusarium wilt caused by *Foc* TR4 in India was initially estimated to be between 2% and 26.6%, a recent survey revealed that the incidence was more than 50% in both Bihar and Uttar Pradesh, where the Cavendish banana is grown on more than one lakh (100,000) hectares [3]. Unexpectedly, no single effective control method has been identified so far except prevention, through the use of pathogen-free tissue culture plants and the adaptation of quarantine strategies [13]. These sanitation practices, however, are mostly followed by commercial plantations and not by subsistence farmers in developing and underdeveloped banana-growing countries.

A better understanding of the fungal genome is required to elucidate the organization of pathogenicity-associated genes/pseudogenes and their virulence mechanisms to develop effective Fusarium wilt control measures. To discriminate the *Foc* races, various molecular methods such as random amplified polymorphic DNA markers [14], restriction fragment length polymorphisms [15], amplified fragment length polymorphism [16], race-specific duplex PCR marker [17], DNA sequence analyses [18], and transcriptomic approaches [19,20,21,22] have been employed. Recently, comparative genomic analysis has been employed to reveal the lineage-specific genomic regions that are responsible for the polyphyletic origin of host specificity [19,23]. Though studies have been established to demarcate each specific race and their molecular mechanisms [17,24], very little is known about the differences between *Foc* races, *Foc* R1, *Foc* STR4 and *Foc* TR4 at the genome levels [20,21,23,25,26,27], and specifically, Indian *Foc* races that have a serious impact on the Cavendish banana [28].

Therefore, the study aims to delineate the differences in the genomes of *Foc* R1, *Foc* STR4, and *Foc* TR4 infecting the Cavendish banana compared to their respective genome references (*Fol* 4287 and *Foc* races), and vice versa, based on putative virulence-associated genes. We believe that a better understanding of the genomes of the various Indian-origin races could aid in the development of race-specific control measures.

## 2. Materials and Methods

### 2.1. Collection of Foc Races

The VCGs of Indian *Foc* races used in the study, VCG0124 (hereafter *Foc* R1), VCG0120 (hereafter *Foc* STR4), and VCG01213/16 (hereafter *Foc* TR4), were obtained from the Plant Pathology Division, ICAR-National Research Centre for Banana, Tiruchirappalli. *Foc* R1 was originally collected from the Cumbum area, Theni District, Tamil Nadu (9°44′05″ N 77°15′02″ E; 482 m MSL), in 2009 by Thangavelu and Mustaffa [29]. *Foc* STR4 was collected from the Burhanpur district (21°19′58″ N 76°12′46″ E 256 m MSL) of Madhya Pradesh [6]. *Foc* TR4 was collected from Barari Village, Katikar District, Bihar (25°14′16″ N 87°01′07″ E, 49 m MSL), in 2015 by Thangavelu et al. [8]. All these strains were isolated from the vascular strand of pseudostems in symptomatic Grand Naine cv. (Cavendish-AAA group). VCG analysis with *nit*-M testers, pathogenicity testing using micro-propagated Grand Naine plantlets, and molecular confirmation with race-specific PCR markers (e.g., [11,17,27]) were performed to reconfirm the virulence and race of the isolates as per standard procedures before the whole genome sequencing. Thangavelu et al. [8] provided a detailed description of these methods elsewhere.

### 2.2. DNA Extraction and Quantification

The genomic DNA of *Foc* strains was extracted from single spore cultures at the late log phase (5–7 days) grown in potato dextrose broth at 25 ± 2 °C under dark conditions without shaking [30]. Mycelia were collected in sterile filter paper and ground into a fine powder in liquid nitrogen with a sterilized pre-cooled pestle and mortar. DNA was extracted from the finely ground mycelia powder using the CTAB method [31] with the use of a phenol-chloroform-isoamyl alcohol mixture, followed by proteinase K and RNase digestion. DNA samples were suspended in TE buffer and used to check the purity as a ratio of A_260_/A_280_ measured by NanoDrop^®^ (M/s. NanoDrop, Wilmington, DE, USA) and agarose gel electrophoresis. The purified DNA was stored at −20 °C until further use.

### 2.3. Genome Sequencing and Assembly

The genomic library was prepared from a sheared DNA fraction of ~300 bp using Illumina paired-end sample preparation kits according to the manufacturer’s instructions. The prepared fungal libraries were sequenced using the Illumina NextSeq^®^ 500 system (M/s. Genotypic, Bengaluru, Karnataka, India) for 150 × 2 cycles, generating ~2.6 million paired-end reads amounting to ~16.18, 29.63 and 17.4 million nucleotides for *Foc* R1, *Foc* STR4 and *Foc* TR4, respectively. The genome data were deposited at NCBI SRA under accession number SRP299372. The standard Illumina pipeline was used to filter the whole genome data. To remove low-quality reads and reads containing adaptor/primer contamination, FASTQ files were further subjected to stringent quality control using the NGS QC Toolkit (v.2.3). The short-read sequences were assembled using MaSuRCA v.3.2.4 [32] with the high-quality sequencing data using k-mer 23, and the gaps were filled using GapFiller44 v.1.11.

### 2.4. Mapping of Foc Genome

In addition to generating *de novo* assemblies, Illumina sequencing reads were mapped to the *Fol* 4287 assembly. Bowtie2 v.2.2.5 was used for the mapping of high-quality filtered reads of *Foc* R1 (GCA_011316005.3), *Foc* STR4 (GCA_016802205.1) and *Foc* TR4 (GCA_014282265.3) against the *Fol* 4287 genome (GCA_000149955.2). Subsequently, only uniquely aligned reads (with mapping quality ≥30 and minimum read depth 10) were considered in this study. Base quality score recalibration and individual variant calls were performed using Genome Analysis Toolkit (GATK) v3.5–0 with the Haplotype Caller module [33] following recommended practices [34].

### 2.5. Identification and Analysis of Variants in the Foc Genome

Genome-wide distribution of DNA polymorphisms was analysed by calculating the frequency of every 100 kb interval on each *Fol* 4287 chromosome. A Circos map (v.0.69–9) was created to visualize the distribution of the SNPs and INDELs related to the *Fol* 4287 chromosome [35]. To identify synonymous and non-synonymous SNPs, and large-effect SNPs and INDELs [36] between the reference and samples genome, samtools v.1.2 and bcftools v.1.2 tools were used and further annotated with SnpEff v.3.4, respectively.

### 2.6. Genome and Functional Annotation

RepeatModeler v.2.0.1, RepeatMasker v.4.0.6 and transposonPSI were used to identify transposable elements and repetitive and low complexity regions in the assembled genome using default settings. Gene prediction was performed on softmasked genomes using Augustus v.3.2, a pipeline for automated training and ab initio gene prediction [37]. The annotations of identified genes were performed with cut-off *E*-values of ≤1 × 10^−5^ and identity ≥30% using the BLAST against the NCBI nr database. Gene ontology (GO) analysis was carried out using UniProt and COG databases [38].

For pathway analysis, protein sequences of *Foc* races were subjected to KEGG database annotation using blastKOALA [39]. A total of 1105 predicted protein sequences were assigned KO identifiers. These assigned KO identifiers were used to map the KEGG database with the help of a KEGG mapper to identify the pathways. To identify the potential pathogenicity-related proteins, a BLASTP search was performed against the Pathogen-Host Interaction database (PHI-base) with a threshold E-value of ≤1 × 10^−5^ [40]. SIX genes in the genome of Indian *Foc* races were defined by using BLAST analysis to compare the assembled genome to SIX genes in the NCBI database [41,42].

### 2.7. Molecular Characterization of Indian VCGs for SIX genes

PCR-based molecular markers and BLAST techniques were used to confirm the presence of SIX genes in the VCGs of Indian *Foc* isolates and their counterparts. Initially, we employed published primers from Taylor et al. [43] for PCR-based diagnosis and found that they were not reproducible for Indian *Foc* VCGs, except for certain SIX genes, as they were intended for *cepae* and other f. sp. of different *Fo* VCGs (Appendix A). Therefore, we designed SIX primers specific to *Foc* based on sequences from different f. sp. of *Fo* available in the NCBI database, regardless of VCGs. To design *Foc* specific SIX primers, a total of 927 homologous SIX gene sequences from 25 f. sp. (Appendix A) were used, and were also subjected to primer-BLAST (https://www.ncbi.nlm.nih.gov/tools/primer-blast, accessed on 24 May 2020) against the target SIX sequences of Indian *Foc* genome assemblies using default parameters. Primers that exclusively hit with *Foc* and genome assemblies were subjected to PCR amplification using standard thermocycling conditions: one cycle of 2 min at 94 °C; 30 cycles of 45 s at 94 °C, 30 sec of annealing temperatures, and 1 min at 72 °C, followed by one cycle of 5 min at 72 °C. Following PCR confirmation, the presence/absence of SIX genes and subgroups was determined using genome BLAST (https://blast.ncbi.nlm.nih.gov/Blast.cgi, accessed on 24 May 2020) against the genome assembly of Indian VCGs and *Fol* 4287 using default parameters, with 927 sequences of SIX homologues used.

Given the relevance of SIX1 genes in defining the pathogenicity of VCGs and banana cultivars, as emphasized by Czislowski et al. [44] and Guo et al. [19], we further investigated the sequence similarity of SIX1a-i genes and their locations in genome assemblies. Further, the aligned SIX1 sequences with maximum similarity were extracted from the genome assemblies, compared and finally subjected to phylogenetic analysis to ascertain the variations between the Indian isolates infecting Cavendish banana and isolates not infecting (i.e., nonpathogenic to) Cavendish banana, e.g., Czislowski et al. [44] and Guo et al. [19]. The extracted and obtained sequences were aligned using the CLUSTAL-W method, and the consensus phylogenetic tree obtained through the maximum likelihood method with 1000 bootstrap resampling was visualized using MEGA-X (v.10.2.5).

## 3. Results and Discussion

The Indian-origin *Foc* VCGs are well-established virulent strains against the Cavendish banana, but they lack a contiguous genome assembly. We assembled the *Foc* genome into chromosome-scale contigs with unplaced and mitochondrial scaffold regions using high-coverage Illumina sequencing. Indian *Foc* races R1 (VCG 0124), STR4 (VCG 0120), and *Foc* TR4 (VCG 01213/16) were confirmed using VCG analysis and *Foc* specific PCR markers [7,11,31].

### 3.1. Genome Sequencing and General Features

The short sequences of *Foc* R1, *Foc* STR4, and *Foc* TR4 were assembled into 88, 85, and 88 scaffolds, respectively, with a total genome size of 61 to 63 Mb and ~48.5% GC (Table 1). Although the genome size of *Foc* TR4 was slightly larger (3.0%, 1.9 Mb) than the reference genome *Fol* 4287, the genome sizes of all three Indian genome assemblies were comparable. Interestingly, we observed a noticeable difference in the *Foc* genome of Indian races, specifically larger genome size, when compared to well-known *Foc* genome assemblies reported from other parts of the world. For instance, the genome of the novel *Foc* TR4 (VCG01213/16) of Indian origin contained 16.3 (*Foc*4_1.0; GCA_000350365.1)—26.4% (FO_II5_V1; GCA_000260195.2) more nucleotides than controls, which translated into a 10.1—16.3 Mb genome size. Similarly, the genome assembly of *Foc* R1 had a 16.8 (*Foc*1_1.0; GCA_000350345.1)—22.5% (ASM593051v1; GCA_005930515.1) larger genome size i.e., 10.1 to 13.5 Mb. In the case of *Foc* STR4, the genome size was 23.9% (14.3 Mb) higher than the C1HIR_9889 (GCA_001696625.1). The difference in genome size between Indian *Foc* VCGs and other VCGs is likely attributable to the fact that we built libraries with larger inserts and obtained more mate-pair information from them, allowing us to connect contigs into scaffolds more easily [19]. Sequenced reads were aligned to the associated assemblies to check the assembly’s integrity; ~97–98% of reads could be aligned to the respective *Foc* assemblies. The high map ratio indicates that the assemblies covered the majority of the genome [19].

The relatedness and completeness of the three *Foc* genomes were determined using the orthologous genes from the Sordariomycetes data set through BUSCO (v.3.0.2). On average, Indian *Foc* VCGs had ~97% (3613) intact, single-copy and complete, 1.8% (65) duplicated, 0.9% (33) fragmented and 0.3% (10) missing orthologous genes out of 3725 target genes. The results showed that the genome assemblies of Indian *Foc* races were robust and complete, and the genome met or exceeded the BUSCO parameters for quality [46], which is in accordance with Asai et al., [23] and Warmington et al. [25].

### 3.2. Transposable Elements in the Foc Genomes

As the presence of DNA repeats and transposable elements (TEs) facilitates changes in genome size and gene expression, and thus phenotypic variation/pathogenicity in *Foc* isolates [45], the variations between the VCGs of Indian *Foc* races were compared with other *Foc* R1 and TR4 genomes. Transposable elements (TEs) were identified using ab initio prediction methods, and TEclass [47] categories were designated as DNA transposon, long interspersed nuclear element (LINE), short interspersed nuclear element (SINE), and retrotransposon with long terminal repeats (LTRs) [48]. Appendix A furnishes a detailed summary of TEs, repeats, and their classification for all VGCs, as well as raw analysis data, which have been stored in the author’s repository [49]. The study accounts for approximately 0.27 to 0.35 Mb of TEs (0.43 and 0.56% of the assemblies) in the Indian *Foc* VCG assemblies (Table 1). When compared to Yingzi [21] and Ma [45], who reported TEs in *Foc* R1 (4.34–5.72%), R4 (4.0%), TR4 (5.22–8.63%) and *Fol* 4287 (3.98), TEs in the Indian isolates were comparatively lower in number and size. VCG0124 (5444) had the most TEs, followed by VCG0120 (5312) and VCG01213/16 (4582), although all were 55.4 to 62.5% less than *Fol* 4287, which had 12,216 TEs (Appendix A). Among individual chromosomes, Chr-3 (541–623) had the most TEs, followed by Chr-1 (497–573), Chr-6 (394–465), and Chr-2 (423–452), while Chr-11 (90–117), Chr-12 (64–94), and Chr-13 (64–94) had the least (55–71) TEs. It is interesting to observe that the highest number of TEs was found in lineage-specific Chr-3 and Chr-6, as well as in Chr-15 (250–288) and Chr-14 (68–219). The total composition of TEs included 20.6–25.1% retroelements, 0.73–0.87% DNA transposons, 21.5–25.9% total interspersed repeats, 32.0–36.5% simple repeats and 40.1–42.2% low complexity regions. The average percentages of deletion and insertion in Indian VCGs were 0.55–0.57% and 0.60–65%, respectively. Deletions were most abundant in Chr-3, 6, 14 and 15, while insertions were commonly richest in Chr-14 and 15, followed by Chr-3, Chr-8 and Chr-1. Previous studies indicated that the greater the number of insertions in the genome, the higher the amounts of chromosomal reorganization, altered expression, and the generation of new proteins [50] and thus variations in genome plasticity [51], pathogenicity [52], host range [53] and evolution [54]. The total number of repeated sequences in Indian VCGs ranged from 14% (VCG01213/16) to 17.8% (VCG0120), which was comparable to *Fol* 4287 (17.5%).

### 3.3. Gene Content in the Foc Genomes

Intact protein-coding genes were identified using a combination of homology-based and ab initio prediction methods. A total of 21,842 intact protein-coding genes were predicted from the consensus gene sets of the *Foc* R1 genome assembly, which was 4.4% (917) higher than the reference *Fol* 4287 in terms of number (Table 1). The total number of protein-coding regions in the reference *Foc* R1 strain NRRL 32,931 was 23,735, which was 8.0% (1893) higher than the *Foc* R1 VCG 0124 from India. The STR4 and TR4 genome assemblies contained 17,118 and 17,745 protein-coding regions, respectively, which were 18.2% and 15.2% less than the reference *Fol* 4287 genome assembly. Although the number of protein-coding regions differed between the gene predicted models of the respective races, ~66.9% of the protein-coding regions in the *Foc* genomes had at least three exons per gene, with an average exon length of 450 bp.

As the number of protein-coding genes across the genome varied, the distribution of the genes within the chromosomes of the different genomes was compared (Figure 1A). Of the total genes, the largest proportions—38.4% (*Foc* STR4) to 42.8% (*Foc* R1)—of genes presented within the Chr-1, 2, 4, and 5 chromosomes, with the maximum of 2534 genes presenting in Chr-4 of *Foc* R1, while 1951 and 2137 genes presented in Chr-1 of STR4 and TR4, respectively. The lowest number of genes was typically observed in Chr-14, where *Foc* R1 encoded only 150, TR4 encoded 268 and STR4 encoded 357 genes. Apart from the 15 chromosomes, the other scaffold held 3.8% to 5.3% of genes of which STR4 had a maximum of 910 genes, followed by *Foc* R1 (820) and TR4 (671).

Similarly, the presence and distribution pattern of rRNAs and tRNAs in the chromosome of each genome was related (Figure 1B,C). The results revealed that the *Foc* R1 contained a maximum of 524 tRNAs and 223 rRNAs, with the highest number of tRNAs present in Chr-12 (100), followed by Chr-11 (64) and Chr-1 (52), and the highest number of rRNAs present in Chr-2 (91). TR4 contained a maximum of 358 tRNAs and 134 rRNAs, with the highest number of tRNAs present in Chr-12 (50) followed by Chr-2 (43) and Chr-11 (40), and the highest number of rRNAs present in Chr-2 (70), Chr-11 (13) and Chr-12 (10). STR4 had a total of 304 tRNAs and 121 rRNAs in the genome, with the highest number of tRNAs in Chr-12 (48) followed by Chr-1 (34), Chr-11 (34) and Chr-2 (33), and the highest number of rRNAs in Chr-2 (38) followed by Chr-11 (9), Chr-12 (9), and Chr-5 (9). We were unable to detect any tRNA or rRNA in Chr-3, Chr-14, or Chr-15 in this study. Surprisingly, the absence of tRNA and rRNA has been linked to the most important lineage-specific chromosomes of the *Foc*, namely Chr-3, Chr-14, and Chr-15, which encode the majority of the virulence-associated genes that determine the pathogenicity of the *Foc* strains [45].

### 3.4. GO and KEGG in the Foc Genome

A total of 8756 (39.5%) of 22,151 protein-coding genes in *Foc* R1, 5384 (28.4%) of 18,946 protein-coding genes in STR4 and 11,220 (57.1%) of 19,651 protein-coding genes from TR4 were annotated based on the Gene Ontology database (http://www.ncbi.nlm.nih.gov/COG/, accessed on 16 June 2020) with a cutoff E-value of 10^−5^. The results are depicted in Figure 2. Of the total of 39.5% annotated proteins in *Foc* R1, 1952 (22.3%) were associated with biological processes, 2969 (33.9%) were associated with cellular processes and signalling, and 3835 (43.8%) were associated with molecular functions (Figure 2A). In the case of TR4, of the 57.1% of proteins annotated, 2626 (23.4%) were associated with biological processes, 4096 (36.5%) with cellular processes and signalling, and 4498 (40.1%) with molecular functions (Figure 2A). In the case of STR4, only 28.4% of proteins were annotated, of which 1325 (24.6%) were associated with biological processes, 1704 (31.6%) with cellular processes and signalling, and 2355 (43.7%) with molecular functions (Figure 2). This study also recorded a large number of protein-coding genes, ranging from 42.9% (TR4) to 71.6% (STR4), which are uncategorized into any of the GO classes and thus considered proteins of uncharacterised functions/features, or potentially genes undergoing rapid evolution and thus displaying high variation that did not match the BLAST result for the given criteria [19]. If we assume that a large number of genes are evolving rapidly, predicting the virulence and pathogenicity of Indian *Foc* TR4 and *Foc* STR4 will be beyond our current or anticipated disease trends, which is an immense reality. This is a concerning result for the banana farmers of northern India because it is a major banana-growing region of the country, accounting for 20.1% of total banana production and 57% of Cavendish banana cultivation. A similar analysis was carried out using the KEGG and Interpro databases to identify the protein-encoding genes involved in the various pathways (Figure 2B). According to the findings, the highest number of 570 genes was identified in the *Foc* TR4 genome, followed by *Foc* R1 (390) and *Foc* STR4 (330). Among the pathway categories, the largest numbers of 116 (STR4) to 202 (TR4) genes belong to the metabolic pathway, followed by the biosynthesis of secondary metabolites (52–105) and biosynthesis of antibiotics (48–87) pathways. The lowest numbers of genes were involved in the fatty acid degradation (9–10), propanoate metabolism (5–8) and tyrosine metabolism (6–8) pathways.

### 3.5. SNPs, InDels and Phylogenetic Relationship

The assembled genome of Indian *Foc* VCGs was scanned with *Fol* 4287 as the reference and a total of 4866 high-quality SNPs were found, with 1502 in *Foc* R1, 1844 in STR4 and 1520 in TR4 (Appendix A; Appendix A) while a total of 139,023 InDels (88,532 insertions and 50,491 deletions) was found of which 53,602 were in *Foc* R1, 33,298 in STR4 and 52,123 in TR4 (Appendix A; Appendix A). The variations in type and positions of the SNPs and InDels between the genomes are presented as a circular Circos plot in Figure 3A. The results indicated that the genes were distributed more densely on longer scaffolds than on shorter ones, which could be attributed to scaffold integrity. SNPs were more common and evenly distributed than InDels. It is worth noting that InDels and SNPs were abundant in lineage-specific chromosomes, *viz*., Chr-3, Chr-6, Chr-14 and Chr-15.

Based on the orthologous genes from the Sordariomycetes data, single-copy genes (3613) from each species were concatenated into a supergene to infer the phylogeny tree, comparing nine published *Fusarium oxysporum* formae speciales genomes. As shown in Figure 3B, the tree indicated a close relationship between the known reference *Fol*, *Foc* and *Fo* genomes, suggesting that they may have descended from a common ancestor. The results showed that the strains of different formae speciales clustered into two clades: clade-1 had four *Fo* members, i.e., *pisi*, *apii*, *melonis* and *sesami*, and clade-2 had *Fol* 4287, *Foc* race-1, TR4 and STR4 with *Fo* f. sp. *physali*, indicating the close phylogenetic relationships among the different formae speciales. Specifically, the four strains of *Foc* were clustered in the same subclade. This hierarchical relationship was in accordance with the race types of *Foc*, as the *Foc* TR4 strains VCG01213/16 and *Foc* NRRL54006 belonged to TR4 and the *Foc* race with VCG0124 belonged to *Foc* race 1, while strain VCG0120 grouped with *Foc*. The divergence of *Foc* races might be caused by several significant race-specific genes [55].

### 3.6. Virulent Genes and Their Functions in the Genome of Foc

To find potential virulence-associated genes, whole-genome BLAST analysis was carried out against the Pathogen-Host Interactions (PHI) gene database (http://www.phi-base.org/, accessed on 16 June 2020), a collection of genes proven to affect the outcome of pathogen-host interactions from fungi, oomycetes and bacteria [41]. After removing the genes that were not related to pathogenicity, we identified 1738 (*Foc* R1) to 2809 (*Foc* STR4) putative virulence-associated genes (Figure 4B). Of the total PHI genes identified in the *Foc* R1, 35.7% (620) were ascribed to biological functions, 22.2% (386) to cellular processes and signalling functions and 42.1% (732) to molecular functions. For *Foc* STR4, 31.8% (892) were ascribed to biological functions, 22.6% (636) to cellular processes and signalling functions and 45.6% (1281) to molecular functions. For *Foc* TR4, 37.3% (755) were ascribed to biological functions, 17.1% (347) to cellular processes and signalling functions and 45.6% (924) to molecular functions.

To identify the unique PHI genes, the functions of virulence-associate genes were compared (Figure 4A). The results revealed that there were 243, 418 and 351 putative virulence-associated genes in *Foc* R1, TR4 and STR4, respectively (Appendix A). The total number of virulence genes reported from this study was comparable to *Foc* R1 and *Foc* TR4, which contained a total of 347 and 348 putative virulence genes, respectively [19]. A total of 46 putative genes were common among the three genomes, while 72 and 30 genes were common between R1 and TR4 and STR4, respectively. Similarly, TR4 shared 16 virulent genes with STR4. A total of 94, 283 and 258 virulence genes were unique to the *Foc* R1, TR4 and STR4 genomes, respectively. The identified PHI genes were subjected to analysis of secreted protein (http://www.cbs.dtu.dk/services/SecretomeP, accessed on 16 June 2020) followed by effector genes (http://www.cbs.dtu.dk/services/SignalP, accessed on 16 June 2020) and were functionally annotated based on BLAST results. The number of effector genes encoded by the genome varied, in which the minimum was 17 genes present in TR4, followed by 20 genes in *Foc* R1 and a maximum of 28 genes in STR4. Similarly, the number of genes involved in increased virulence was 13, 23 and 27 in TR4, STR4 and *Foc* R1, respectively (Figure 4B).

Among the virulence-associated genes, ABC1, kdpB, acrB, oqxA & B and pstB were cellular transporter protein-encoding unigenes present in the *Foc* genome (Appendix A), which are essential for the import of nutrients and export of secondary metabolites [56]. ABC is one of the largest gene families in the ATP-binding cassette transporters superfamily, which hydrolyse ATP to transport a wide range of substrates across biological membranes. ABC genes can be either importers or exporters in the cell membrane depending on the direction of transportation relative to the cytoplasm [57], and have been divided into seven families (A–G) in fungi [58]. Transgenic expression of ABC genes in *Arabidopsis thaliana* indicated that the ABC transporter gene is required for its transportation of salicylic acid (SA), fungicide resistance, mycelial growth and pathogenicity. SA is a critical plant defense hormone, which contributes to plant defense against a wide range of pathogens with biotrophic and hemibiotrophic lifestyles [59]. A study estimated that *Fusarium* sp. contained ~45–61 ABC transporter, which was found to provide tolerance to a wide array of antifungal compounds generated by a broad host range [45].

kdpB is part of the high-affinity P-type ATP-driven transporter which catalyses the hydrolysis of ATP coupled with the electrogenic transport of potassium into the cytoplasm. Expression of kdp in *Staphylococcus aureus, Yersinia pestis*, mycobacteria etc. [60,61] has been found to connect with the regulation of several virulence genes during pathogenesis and also contributes to survival under a variety of stressful conditions [62]. AcrB is a resistance-nodulation-cell division family of efflux pumps that plays a major role in a multidrug efflux system by facilitating intrinsic resistance mechanisms. Experimental results demonstrated that the presence of AcrB and overexpression of an efflux system significantly increases virulence in *Caenorhabditis elegans* [63] and *Klebsiella pneumoniae* through the expression of OqxA and OqxB efflux pump [64]. It is hypothesised that an increase in mutation frequency enables the rapid evolution of high-level resistance via the accumulation of point mutations.

The Pst (phosphate transport) system is involved in transporting *P_i_*, which is typically composed of four types. Specifically, cytosolic PstB energizes the release of free *P_i_* in the cytoplasm through ATP hydrolysis function in the transmembrane channel. Peirs et al. [65] reported that Pst systems are important for intracellular survival in the host and play a crucial role in virulence [66].

Additionally, in the *Foc* TR4 genome, Chr-4 encodes an intracellular signalling system, known as two-component regulatory systems (PXO_04659), which regulate pathogenesis and biological processes as very similar in *Xanthomonas axonopodis* pv. *citri* was identified. A study showed that a knockout strain lacking the PXO_04659 gene exhibited a drastic reduction in virulence relative to the wild-type [67]. The enhanced virulence generally observed in *Foc* TR4 may be attributed to detoxifying regulator mechanisms involved in increasing extracellular polysaccharide synthesis, tolerance to reactive oxygen species, and iron homeostasis, either alone or in cooperation with other regulatory factors [67].

### 3.7. The Organisation of Secreted in Xylem Protein-Encoding Genes in Foc Genome

BLASTx analysis and PCR based primers were employed to characterise the presence or absence of secreted in xylem SIX genes (SIX1-SIX14) in Indian *Foc* isolates [42]. The presence of SIX genes and their subgroups was recognized in this study if the VCG showed both PCR amplification to a molecular marker and a BLAST hit with >80% similarity with *Foc*/*Fol*; otherwise, it was regarded as absent (Appendix A). The difference in PCR results with BLAST hits or previously reported primers (e.g., [42], [44]) may, however, be attributed to sequence variation in the genome or optimal annealing temperature of the primer [28]. As we manually checked the sequencing depth and synteny relationship near the SIX regions, we ruled out assembly errors and assumed that the difference was most likely due to strain variations and other unknown factors [19].

Genome BLAST analysis including 927 SIX query sequences of different formae speciales reveals (Appendix A) that VCG0120 (1039) had the highest number of hits followed by VCG01213/16 (929) and VCG0124 (653). *Foc* VCG0124 had the highest number of hits with *Fo* f. sp *canariensis* (152), followed by *Foc* (91) and *Fo* f. sp. *pisi* (42). *Foc* VCG0120 had the highest number of hits with *Foc* (201), *Fo* f. sp. *canariensis* (152) and *Fo* f. sp. *palmarum* (73). In the case of *Foc* VCG01213/16, the maximum number of BLAST hits registered with *Foc* (153), *Fo* f. sp. *canariensis* (152), and *Fo* f. sp. *palmarum* (73). Similarly, the maximum number of hits was compared based on the number of SIX hits. *Foc* VCG0124 had a maximum number of hits with SIX1 (135) followed by SIX7 (102) and SIX6 genes, while VCG0120 recorded a maximum of 284 hits with SIX8 followed by SIX1 (136), SIX13 (110), SIX7 (102) and SIX6 (99). *Foc* TR4 had a maximum number of hits with SIX8 (223) followed by SIX1 (139), SIX6 (115) and SIX7 (102). Although the analysis resulted in SIX genes that were not specific or reported previously in *Foc,* this study only considered the presence of the SIX genes and subgroups if the PCR result was positive. Therefore, the following discussion on the presence and absence of SIX genes is based on the concurrent results of both PCR and BLAST.

Based on the variation in the SIX gene profiles of *Foc* races reported previously [17,42], it was found in the present study that SIX11 and SIX14 were absent in all three Indian *Foc* genomes (Table 2). Among the SIX genes, SIX1, SIX4, SIX6 and SIX13 were present in the *Foc* R1 genome whereas SIX8 and SIX9 were absent. We found differential results with SIX2, SIX3, SIX5, SIX7, SIX10 and SIX12 between BLASTx and PCR markers. The study identified that two homologous copies of SIX1 and SIX13 and a single copy of all the remaining SIX genes were present in Chr-14 and Chr-13. A copy of the SIX1 gene (alternatively *AVR3*) present in Chr-14 had sequence polymorphism at amino acid positions 205–214 (by deletion), which could explain why *Foc* R1 infects Cavendish banana so unusually in India [68]. This corroborates the findings of Li et al., [69], who found that deletion and complementation of SIX1 in *Fusarium oxysporum* f. sp. *conglutinans* resulted in increased virulence against cabbage.

In the case of STR4, SIX1, SIX2, SIX7, SIX8, SIX9 and SIX13 were present and SIX12 was absent. The presence of SIX3, SIX4, SIX5, SIX6, and SIX10 was detected by genome BLASTx but not by PCR markers. The genome contained two homologous copies of SIX1 and SIX9 in Chr-14, as well as 13 homologous copies of SIX8 distributed across Chr-2 (1), Chr-3 (3), Chr-6 (2), Chr-7 (1) and Chr-14 (6) and a copy of SIX8b in Chr-14. The strain of STR4 possessed a homolog of SIX-6, an effector gene located in the Chr-14 between 4258 and 4987 nt which facilitates colonisation of the host, specifically by suppressing I-2-mediated cell death [70].

SIX1, SIX2, SIX8, SIX9 and SIX13 have been found to present in the *Foc* TR4 genome alongside the SIX8a, which is specific to TR4 [44]. Our BLASTx analysis revealed the presence of SIX3, SIX4, SIX5, SIX6, SIX7, SIX10, and SIX12 in the genome, but PCR primers failed to generate amplicons. Similarly to *Foc* R1 and STR4, TR4 has two homologous SIX1 in Chr-14 while 13 homologous copies of SIX8 are distributed in Chr-2, Chr-3, Chr-6, and Chr-14 without any sequence variations, which corroborates the report of Czislowski et al. [44] especially in the VCG 01213/16. They reported that the presence of multiple copies of *Foc* TR4 specific SIX homologues is usually acquired through a horizontal transfer event from other *Foc* formae speciales. In general, SIX8a is present in all race 4 isolates, whereas SIX8b is present in all subtropical race 4 isolates, but has yet to be detected in *Foc* TR4 isolates [71]. However, copies of two SIX8 homologues, SIX8a and SIX8b, were found in the genomes of *Foc* TR4 (in Chr-14) without sequence variation, which is a major difference noted in this study. As a result of the SIX8b effector gene, the *Foc* TR4 of India can infect Cavendish bananas grown in the subtropical region as well, and this confirms the fact that the *Foc* TR4 was isolated from the Katihar district of Bihar which is within the subtropical climate zone of India.

Genome BLASTx analysis revealed that all SIX1a-i homologues (compared to Czislowski et al. [32] and Guo et al. [33]) were present in the pathogenic lineage of Chr-14 in VCG0124 (CM027182.2), VCG0120 (CM028826.1), and VCG01213/16 (CM026317.2) with 82% to 84% similarity, 79 to 92 bp mismatches, and 12 to 15 bp gaps (Appendix A). Even though the SIX1 sequences of Elizabeth and Guo differ slightly, we were able to obtain a maximum of >80% similarity with the genome of Indian *Foc* VCGs infecting the Cavendish banana. According to Czislowski et al. [44], SIX1a, d, f-i have been found to present in VCG0124, 0120 and 01213/16, of which SIX1d and f belong to VCG0124, SIX1g belongs to VCG0120, and SIX1a, h and i are found in VCG01213/16, alongside SIX1b in VCG0121. In contrast, the BLAST results of 927 SIX gene sequences showed that Indian VCGs had the best match with SIX1a, b, and f, where SIX1a and SIX1b are from *Fo* f. sp. *physali*, and SIX1f is from *Foc*. The study emphasis that describing the presence of SIX1 subgroups in VCGs or lineage/horizontal gene transfer mechanisms between race/f. sp is very difficult to comprehend solely on sequence similarity without PCR confirmation, as we do not have VCG/SIX1 specific primers to investigate at the moment. Multiple sequence alignment of extracted SIX1 gene segments revealed significant variations between the VCGs of Indian isolates and those isolated by Czislowski et al. [44] and Guo et al. [19], specifically aligned positions 207–230 and 535–552 (Figure 5A). This variation could be interpreted as the principal difference of Cavendish infecting *Foc* VCGs of India. Furthermore, maximum likelihood phylogenetic analysis revealed that Indian *Foc* isolates were found to be a distinct lineage with a separate clade (Figure 5B) when compared to the SIX1a-i sequences of Czislowski et al. [44] and Guo et al. [19]. This finding supports our hypothesis that *Foc* VCGs of Indian races evolved as a separate lineage with various combinations of SIX genes and subgroups or effector genes.

## 4. Conclusions

When compared to *Foc* genome assemblies reported from other parts of the world, this comparative whole-genome sequence analysis revealed significant differences in Indian *Foc* races in terms of genome size and protein-coding regions. Furthermore, annotation of the TR4 and STR4 genomes revealed that the genome contains 42.9–71.6% protein-coding genes with unknown functions/features or genes undergoing rapid evolution, emphasizing the importance of early detection methods. The current study also explained the mechanisms of the major pathogenicity-related protein families that are involved in increasing the pathogenicity and virulence of organisms. Moreover, the study established the presence of both SIX8a and SIX8b with multiple copies of SIX1, distinguishing the examined TR4 genome from other references. Based on our findings, we believe that a new accurate and early diagnosis procedure is required to identify rapidly evolving *Foc* races of India to sustain the country’s banana production in these regions. In addition, the variable sequence/genome regions discovered in the *Foc* TR4 of India will be useful in future to develop race-specific molecular markers targeting SIX genes and TEs.

## Figures and Tables

**Figure 1 jof-07-00717-f001:**
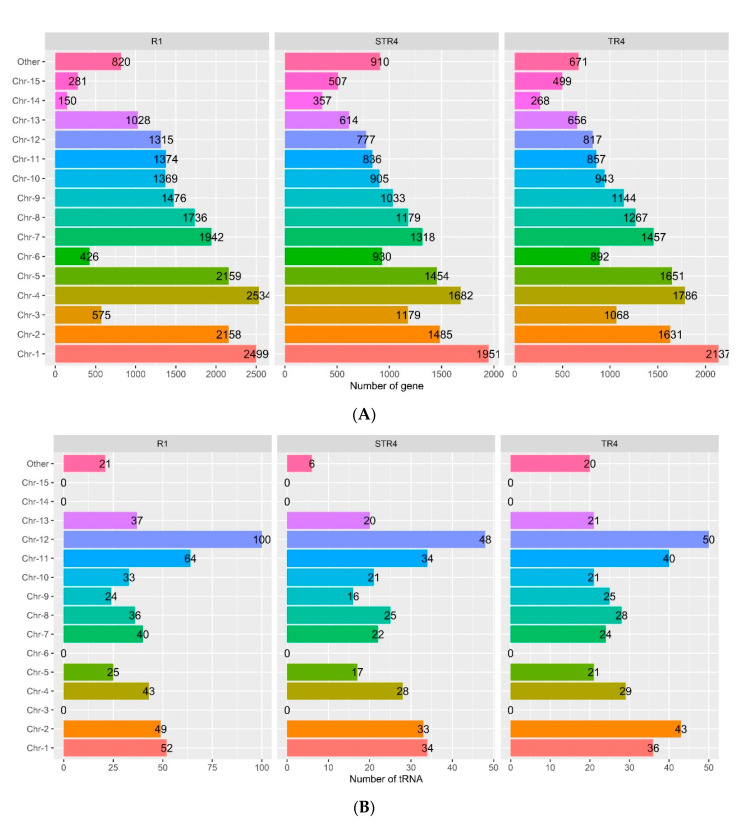
Comparison of the distribution pattern of protein-coding genes (**A**), tRNA (**B**) and rRNA (**C**) in the Indian VCGs of *Foc* genome assemblies. The colour in the bar chart represents the respective chromosome and the value in the bar are the number of occurrences.

**Figure 2 jof-07-00717-f002:**
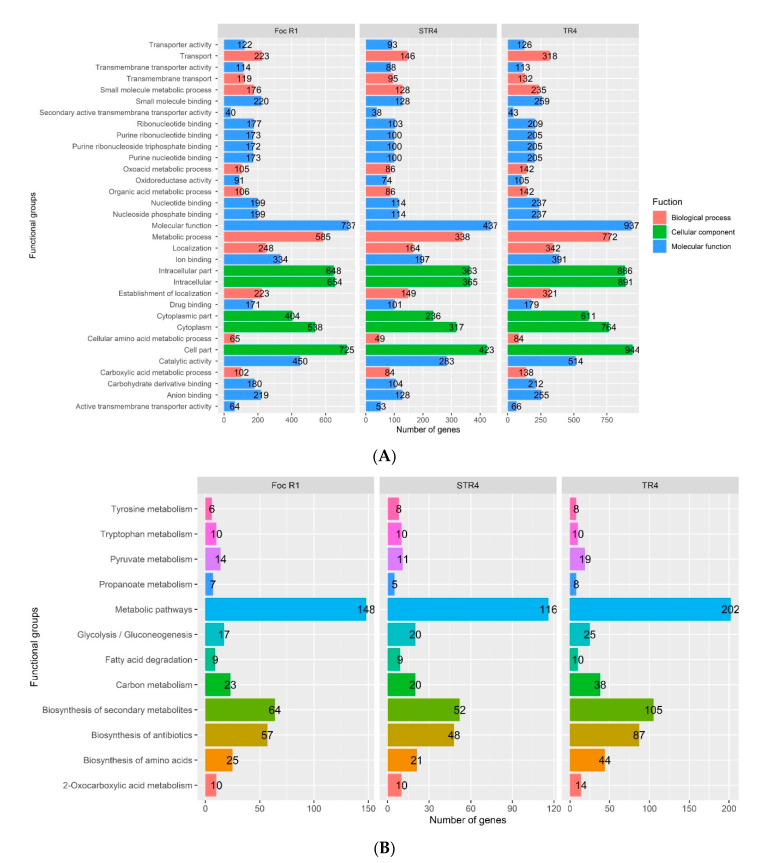
Functional annotation of the Indian VCGs of *Foc* genome assemblies based on COG (**A**) and KEGG (**B**). The colour in the bar represents the respective functional groups in the *y*-axis and value represents the number of genes identified from the respective functional groups.

**Figure 3 jof-07-00717-f003:**
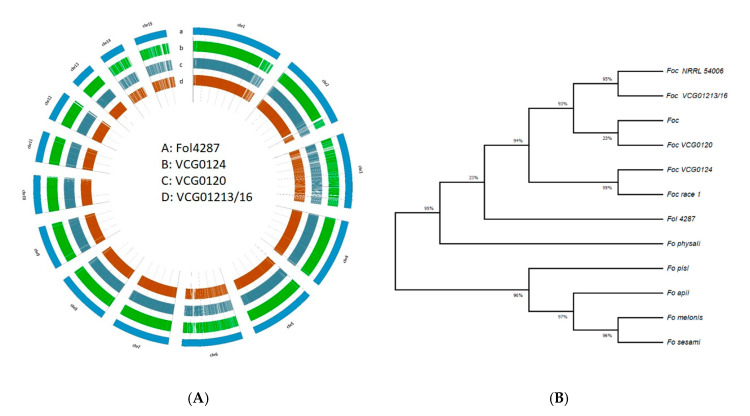
Whole-genome comparison of three Indian *Foc* races and phylogenetic analysis. (**A**) Whole-genome variations between *Foc* R1, *Foc* STR4 and *Foc* TR4 and reference *Fol* 4287, where a: *Fol* 4287; b–d: assembled *Foc* genome of R1, STR4 & TR4; and e: reference genome of *Foc* R1 (ASM593051v1). (**B**) Phylogenetic analysis of nine *Fo* strains from different f. sp resulting in two clustering clades, using MegaX 10.2.5 with the maximum-likelihood method.

**Figure 4 jof-07-00717-f004:**
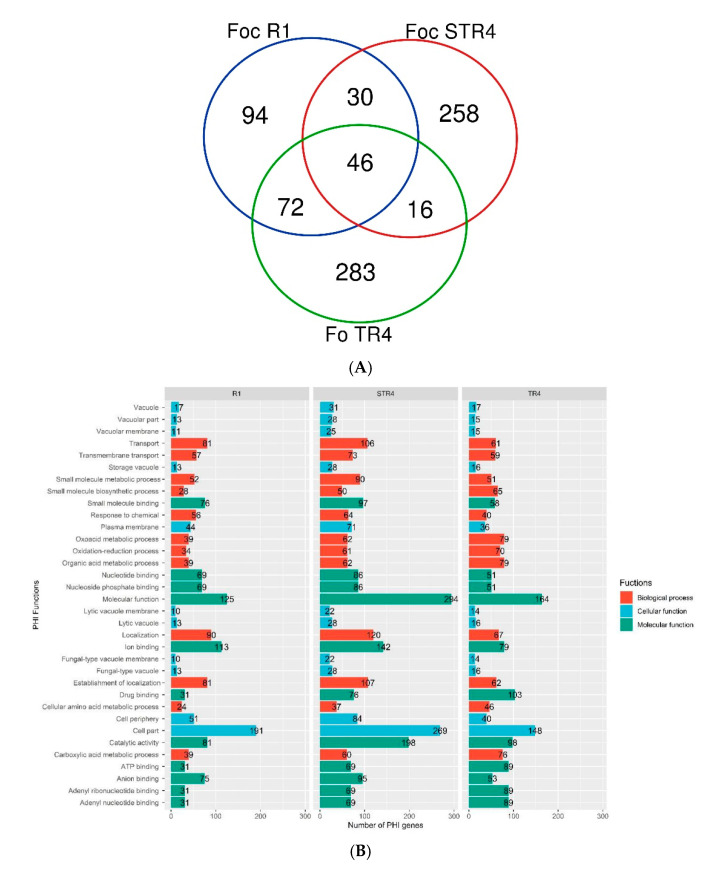
(**A**) Venn diagram of the putative virulence-associated genes identified in the comparisons of *Foc* R1 vs. *Foc* STR4 vs. *Foc* TR4. (**B**) Number and functions of PHI genes present in the Indian *Foc* races.

**Figure 5 jof-07-00717-f005:**
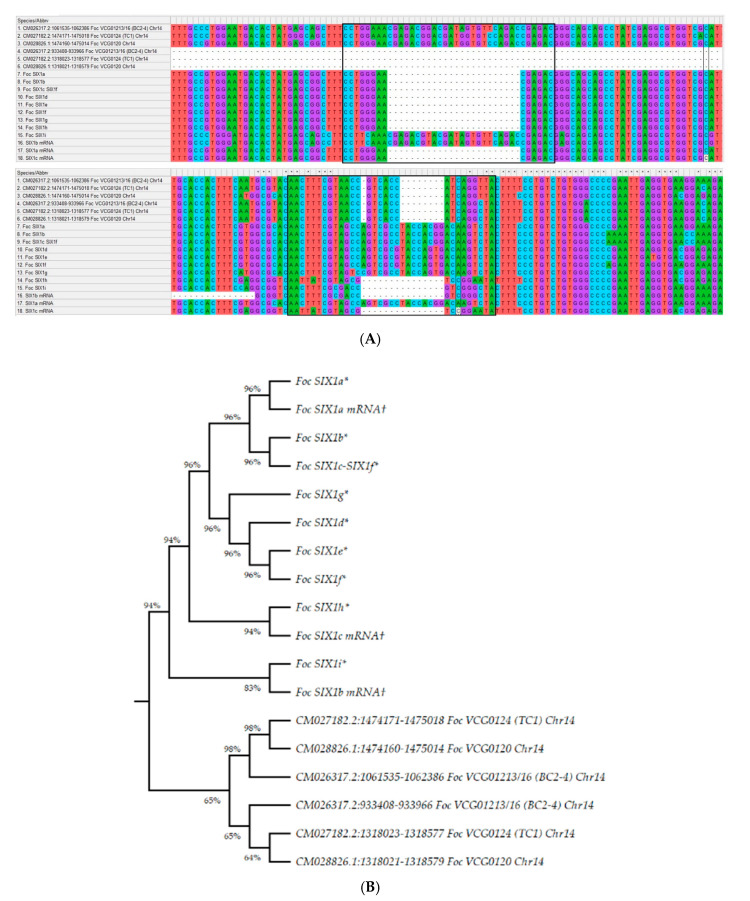
Comparison of SIX1a-i genes of *Foc* VCGs infecting the Cavendish banana with the sequences of Czislowski et al. and Guo et al. (**A**); phylogenetic analysis based on SIX1a-i using MegaX 10.2.5 with the maximum-likelihood method (**B**). Numbers represent bootstrap values from 1000 replicates. The SIX1a-i sequences obtained from Czislowski et al. [44] and Guo et al. [19] are denoted as * and †, respectively, and the sequences of Indian VCGs were obtained from assembled genomes of *Foc* R1 VGC0124 (TC1): GCA_011316005.3; *Foc* STR4 VCG0120: GCA_016802205.1; and *Foc* TR4 VGC01213/16 (BC2-4): GCA_014282265.3.

**Table 1 jof-07-00717-t001:** Summary of genomic and predicted features of reference and assembled genomes of *Fol* 4287 and Indian *Foc* races.

Characteristics	*Fol* 4287	*Foc* R1 (VCG0124)	*Foc* STR4 (VCG0120)	*Foc* TR4 (VCG01213/16)
Genome size (bp)	61,386,934	61,471,473	61,380,681	63,220,715
No. of scaffolds (Count)	114	88	85	88
No. of contig (Count)	1362	1371	1362	6560
Scaffold N_50_ (Mb)	1,976,106	4,781,098	4,589,962	4,589,937
Maximum contig length	6,854,980	8,732,082	6,853,100	7,450,211
Minimum contig length	900	898	888	886
Average contig length	698,542	884,718	722,130	718,421
Median contig length	15,960	17,299	14,354	15,960
GC content	48.4	48.5	48.4	48.5
Protein count	27,347	22,151	18,946	19,651
Total genes (Augustus)	20,925	21,842	17,118	17,745
BUSCO (%)	96.4	98.4	96.9	98.9
Secretome Genes	1847	1949	1870	2330
tRNA	320	524	304	358
rRNA	125	223	121	134
ORF	892,698	1,149,990	891,736	926,420
Repeats (bp)	10,770,810 ^‡^	10,755,517	10,901,072	8,844,756
Total transposable elements (bp)	2,446,574 ^‡^	346,254	342,987	273,767
Retroelements	433,505 ^‡^	87,052	85,124	56,365
DNA transposons	506,130 ^‡^	2527	2591	2391
Total interspersed repeats	342,368 ^‡^	89,579	87,787	58,756
Simple repeats		115,058	110,011	99,987
Low complexity region		141,331	144,912	114,926

*Fol* 4287: GCA_000149955; *Foc* R1 (TC1): GCA_011316005.3; *Foc* STR4: GCA_016802205.1; and *Foc* TR4 (BC2−4): GCA_014282265.3, are genome assembly numbers of the respective strains available in the NCBI database. ^‡^ According to Ma et al. [45].

**Table 2 jof-07-00717-t002:** Profile and inter-comparison of SIX genes in the reference *Fol* 4287 and assembled *Foc* races of Indian origin.

Accession	SIX1	SIX2	SIX3	SIX4	SIX5	SIX6	SIX7	SIX8	SIX9	SIX10	SIX11	SIX12	SIX13	SIX14
*F**ol* 4287	+	+	+	−	+	+	−	+	+	+	−	−	+	−
*Foc* VCG0124 (TC1 ‡)	+	−	−	+	−	+	−	−	−	−	−	−	+	−
*Foc* VCG0124 (TC1 *)	+	++	+	++	++	++	+	−	−	+	−	+	+	−
*Foc* VCG0120 STR4 ‡	+	+	−	−	−	−	+	+	+	−	−	−	+	−
*Foc* VCG0120 STR4 *	+	++	+	++	++	++	+	+	++	+	−	−	+	−
*Foc* VCG01213/16 (BC2-4 ‡)	+	+	−	−	−	−	−	+	+	−	−	−	+	−
*Foc* VCG01213/16 (BC2-4 *)	+	++	+	++	++	+	+	+	++	+	−	++	+	−

* Represents SIX in the genome of *Foc*; where +: present; −; absent in both PCR and BLAST; ++: presence of SIX other than *Foc* f. sp. genome where *Foc* TC1: GCA_011316005.3; *Foc* STR4: GCA_016802205.1; and *Foc* BC2-4: GCA_014282265.3. ^‡^ Results of SIX specific PCR marker.

## Data Availability

The data on transposable elements [49] and gene structure annotations and sequences used for genome BLAST and phylogenetic analysis [48] have been deposited in the Mendeley Data repository. The assembled short-read genome sequences of *Foc* Race 1VCG0124, *Foc* STR4 VCG0120 and *Foc* TR4 (VCG 01213/16) infesting Cavendish banana have been deposited at DDBJ/EMBL/GenBank under accession number GCA_011316005.3, GCA_016802205.1 and GCA_014282265.3 respectively. Raw Illumina sequencing data of Indian *Foc* races are available in the NCBI Sequence Reads Archive (SRA) under the accession number SRP299372. All the data generated or analysed during this study are included in this published article.

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
