# Peer review of "Comparative Whole-Genome Sequence Analyses of Fusarium Wilt Pathogen (Foc R1, STR4 and TR4) Infecting Cavendish (AAA) Bananas in India, with a Special Emphasis on Pathogenicity Mechanisms"

_jof, 2021, doi:10.3390/jof7090717_

Round 1
Reviewer 1 Report
Dear authors,
Most of my previous comments and requests have been addressed and discussed. Some additional analyses were done as requested, which add value to this MN.
Sincerely,
Author Response
All the suggestions given by reviewer 1 are carried out
Reviewer 2 Report
This research article follows up on the resource announcement by Thangavelu et al, 2021 in Plant disease. It reports datasets and comparative genome analyses related to Foc races in India which makes it a valuable resource for publication, thus providing insight on Fusarium genomics.
However, I have major concerns regarding some dataset produced (see below).
Major points:
1/ Genome sizes and SIX genes presence/absence
First, I was very intrigued by the difference of genome size in the TR4 Indian strain compared to previously reported strains. All published ones until now has been comparable in size (between 46 and 48 Mb). This is obviously a critical element to be resolved in this study as it can raise some suspicion on the accuracy of produce datasets.
From authors’ observations, it is not linked to the transposable elements, but they don’t give much clues. my advice would have been:
- Compare the results of the assembly with results from another assembler such as Spades or Abyss just to check the assembly size and eliminate possible methodological flaw.
- If nothing found, map the contigs/scaffolds on the TR4 assembly from Warmington et al, 2019 which is the most contiguous assembly so far and check the content of contigs/scaffolds that do not match their 15 contigs.
- Alternatively, cluster the 14,472 protein-coding genes from Warmington et al, 2019 and the 19,651 protein-coding genes predicted here trying to define the main gene families for those 5000 additional genes?
Second, SIX genes presence /absence compared to current knowledge is also puzzling.
The presence on SIX13 in STR4 is not expected according to Czislowski et al., [35] while SIX2 would be expected in both STR4 and TR4 as reported by Czislowski et al., [35] and Guo at al, [36]. However, they are absent in this study.
As I am experienced with genome analyses, I made a quick check with the India TR4 data.
- I downloaded the SRR13311628 Reads on NCBI SRA entitled “Draft genome of Fusarium oxysporum f. sp. cubense Tropical Race 4 VCG01213/16” linked to the SRP299372 accession number indicated in data availability (it should correspond to Fo TR4).
- I produced a draft assembly using Masurca v4.0.3 (default parameters) and obtained 1,980 contigs for a genome size of 48,920,303 bp.
- I used blast with SIX2 gene reported by Czislowski et al., 2017 as a query to the produced assembly and found one hit
Query= KX435003.1 Fusarium oxysporum f. cubense strain BRIP44012 secreted in xylem 2 (SIX2) gene, complete cds
Hit Score = 1269 bits (687), Expect = 0.0
Identities = 695/699 (99%), Gaps = 0/699 (0%)
I therefore can’t figure out how the authors could produce a 60Mbp assembly with no hit with SIX2. (I checked on GCA_014282265.3 and it is not present indeed.)
I wish I made a mistake in my quick verification and I am happy if the authors can point it out because otherwise, based on gathered evidence, I would have to reject the manuscript.
The remaining comments are secondary and to be addressed only if the point 1 is solved.
2/The introduction is quite comprehensive about the context of Fusarium wilt but when the state of the art in Fusarium genomics is not properly introduced.
First, “Recently, comparative genomic analysis has been employed (…) host specificity [4]”. The reference cited is from 1962, which must be an error.
Then,
L83: the authors states that little is known about the differences of Foc races, Foc R1, Foc STR4 and Fo TR4 at the genome levels without citing any existing genomic resources. There is a number of studies that produced reference genomes and that are missing here. Moreover, the study by Guo et al., [36], is not mentioned here whereas they performed a genome and transcriptome comparison between Foc R1 and Fo TR4. What about Qin et al, G3 2017 ?
Also, it exists important information comparing SIX genes across Foc races by Czislowski et al., [35] which is at the heart of the analyses of the paper. This group subsequently published in Carvalhais et al [19] a diagnostic method (with an explicit title: Diagnostics of Banana Fusarium Wilt Targeting Secreted-in-Xylem Genes) not mentioned in the introduction when listing the various molecular methods used to discriminate Foc races. Those references are present in the list of references but should have been introduced better and earlier.
Minor points:
L20: Please rephrase “In addition to the GO, SNPs, and InDels” by something like “while comparing the general features of the genome sequences (e.g. GO, SNPs, and InDels)”. what about transposable elements?
L24: I would remove “when compared to the reference genome” as this is not clear what is the reference genome across several races. Or replace by “when compared to their respective reference genomes”
L30: where does come the evidence that banana is the fifth most vital horticultural fruit? A reference would be needed.
L44 : Fo TR4 or Foc TR4?
L189: I am not sure why personal communication with the authors was necessary to obtain the sequences? Sequences from the article were deposited at Genbank (KX434991- KX434999)
L200: The section 3.1. Genome Sequencing and General Features for clarity should be divided with sub heading for assembly size, TE content and Gene content.
L496: “the SIX1 sequences of Elizabeth and Guo” I understand that data are coming from personal communication but there might be a better way to refer to it like the reference. The same applies to leaf names in the tree of the Figure 6.
Figure 6: It would be useful for further analyses to add as supplementary data as FASTA format all the sequences used to produce the phylogenetic tree of the Figure 6. For some reasons, the gene annotation has not been deposited at NCBI or elsewhere for download.
Author Response
Files are attached. All the suggestions are carried out

Round 2
Reviewer 2 Report
First of all, I thank you the authors for the elaborated and detailed answers to my previous review.
On the genome size, I knew the existing size variation that existed in Fusarium but was surprised by the one reported for TR4 specifically. In the discussion, it is specified “The difference in genome size between Indian Foc VCGs and other VCGs is likely attributable to the fact that we built more libraries with large inserts and obtained more mate-pair information from them, allowing us to connect contigs into scaffolds more easily.”
I doubt that for the TR4 strain the size varies due to advancements in sequencing technology, methods of assembly and mapping techniques. Actually, a 48Mb size was reported with Pacbio data and we generated illumina and Oxford Nanopore data (unpublished) with the same size using various assemblers. Moreover, the size of the 15 chromosomes is already 59.8Mb so the 3 Mt contigs does not explain the difference. somehow, a lot of additional check were performed based on my suggestions and as pointed out, NBCI accepted this resource. There as still unresolved questions that may be revealed in a next study on FOC genomes.
Regarding the SIX2 gene detection, I did not completely understand the explanation since with the assembly I made with authors’s data comprised the sequence with Czislowski et al. 2017 but was found absent from the assembly submitted by the authors to NCBI. Whatever the reference used for the assembly, it should not change the sequence content. Somehow, results were updated and profiles are now more consistent with previous TR4 reports.
Figure 5. the phylogenetic tree still contains the labels Elizabeth and Guo instead of allelic variant names.
Author Response
We have carried out the corrections as per the reviewer suggestions

This manuscript is a resubmission of an earlier submission. The following is a list of the peer review reports and author responses from that submission.
Round 1
Reviewer 1 Report
Dear authors,
My comments are in the enclosed file.

Reviewer 2 Report
Sequencing and bioinformatic analysis are meticulous. The work, although instructive, is essentially descriptive and doesn’t seem to me to report much innovation. Nevertheless, the work adds informations on the structure of Indian Foc isolates genomes that could be useful for scientific community.
The report was difficult to follow because of the language. . I suggest the authors to have a native English speaker edit their manuscript before re-submitting. In the text attached I highlight words/sentences that sound to me as grammar mistakes, but correction must be performed by a native English speaker.
Line 14-21: I would move generic information in the introduction so to leave more space in the abstract to describe what has been done in this study.
Line 162-164: To assemble the Foc genome into 15 chromosome-based contigs the procedure has been necessarily reference guided on a F.oxysporum genome with known chromosomes,probably Fol4287.
Line 201-202 (and elsewhere in the manuscript): Analysis of chromosome organization has not been performed in lab (pulsed-field electrophoresis), thus talking of putative chromosome is more correct.
Some other comments in the text.
At least in my PDF version, all Figures and Tables have low resolution.
Reviewer 3 Report
Evaluation and comments to the manuscript ID jof-1224111 entitled “Comparative whole genome sequence analyses of Fusarium wilt pathogen (Foc R1, STR4 and TR4) infecting Cavendish (AAA) bananas in India- a special emphasis on pathogenicity mechanisms”.
Authors: Thangavelu Raman et al.
In my opinion, it is an interesting and well-written work. Overall, the experimental design and data analysis are appropriate and the introduction is correct.
My suggestions:
1) Lns 86-90, a new paragraph, the new line: “Therefore, the study aims ...........measure of the disease in banana.”
2) Figs. 1, 2, 4 and 5 - not readable, improve quality
3) Why is "Figure-2, Figure-2a etc." in red in the text?
Round 2
Reviewer 1 Report
The authors have improved the writing of text, and changed some tables and figures as requested. However, some important points were not taken into account, notably my comments on the SIX gene distribution. It seems that the authors preferred to submit a quick revision instead of doing the necessary supplemntal analyses. So in my opinion, this MN has not been sufficiently improved.